# High-Performance Five-Phase Axial Flux Permanent Magnet Generator for Small-Scale Vertical Axis Wind Turbine

**Ketut Wirtayasa, Chun-Yu Hsiao *** and **Nien-Che Yang**

Department of Electrical Engineering, National Taiwan University of Science and Technology, Taipei 106335, Taiwan; ktwirtayasa@yahoo.co.id (K.W.); ncyang@mail.ntust.edu.tw (N.-C.Y.)
* Correspondence: yuhsiao@mail.ntust.edu.tw

**Abstract:** Commonly, electrical energy is generated by using non-renewable energy such as natural gas, coal, and oil. As electrical energy is a basic asset for the development of a region, its utilization is increasing every year, which causes the existence of non-renewable energy to decrease every year. This issue is becoming a serious concern all over the world, which encourages every country to harness energy from renewable energy. Wind energy is a promising candidate for generating electricity today. In wind turbine generation, a three-phase generator is usually used. Along with the rapid development of power electronic devices and efforts to improve generator performances, the use of a multiphase system is considered important for harnessing energy from the wind more efficiently. In this study, a five-phase system is proposed to upgrade the output power and power density of the most qualified AFPMG in the previous study. The Taguchi optimization method is employed to obtain the lowest total harmonic distortion (THD) of the on-load voltage waveform. In addition to the Taguchi method, an Artificial Neural Network (ANN) is also employed to compare the results from the Taguchi method and the results are proven to have an excellent relationship. The data processed for Taguchi and ANN methods are strongly helped by using the finite element method from the Ansys Maxwell software. The performances of the proposed five-phase axial flux permanent magnet generator (FP-AFPMG) show good improvement, especially in THD, ripple torque, and ripple in the rectified voltage.

**Keywords:** five-phase axial flux permanent magnet generator (FP-AFPMG); taguchi method; artificial neural network (ANN) method; generator performances

## 1. Introduction

Electrical energy is a basic asset for the development of a region. Commonly, non-renewable energy such as natural gas, coal, and oil is used as fuel to generate electrical energy [1]. Due to the scarcity and rising prices of non-renewable energy resources today, it is important to utilize other types of resources to generate electricity. One of the promising renewable energies that can be used to generate electricity is wind energy because of its several advantages [2,3].

In a traditional wind generation system, to generate electricity a three-phase generator is widely used. Along with the rapid development of power electronic devices [4] and efforts to improve generator performances, the use of a multiphase system is considered important for harnessing energy from the wind more efficiently. A five-phase motor or generator is categorized as a multiphase system because of its phase number equal to four or higher as well as its arrangement in n-phase symmetrical stator winding configurations [5]. Some of the studies dealing with five-phase permanent magnet synchronous generators are highlighted below. In [6], Kumar et al. proposed a permanent magnet generator for wind power application. The generator was designed with a dual stator where both stators accommodated five-phase windings. The generator is also claimed to be capable of forming eight magnetic poles using only four actual permanent magnet poles on both surfaces of

the rotor. The performance results show that the proposed generator has a higher power density than the two conventional five-phase generators. The advantage and application of five-phase permanent magnet machines are discussed in [7]. The author designed a five-phase generator and its power electronics system. The analysis results show that the five-phase permanent magnet generator is able to reduce the ripple torque, reduce the size of the capacitor, improve power density, and improve the fault tolerance when compared to a three-phase permanent magnet generator. The authors also discussed that the five-phase permanent magnet generator can be applied in a wind turbine system and an aerospace application. In [8], they proposed an analytical method to obtain the optimal performances of a five-phase permanent magnet synchronous generator in an improvement time duration when compared to the finite element method (FEM). The authors used a multiphase system in designing the generator for a direct-drive wind turbine because the multiphase system has the superiority of reducing the dimension of the converter, increasing the power density, increasing the reliability, and increasing the fault tolerance capability when compared to the three-phase system. Due to the increasing interest in utilizing renewable energy caused by the scarcity of non-renewable energy, the authors in [9] designed a generator with a five-phase system that can be applied in a gearless small-scale wind turbine. The multiple optimizations were carried out on its geometry and were analyzed with the two-dimensional finite element methods (FEM). By using a five-phase system, the simulation results showed that the generator voltage ripple was greatly reduced, leading to an ideal rectifier waveform.

Based on the advantages of the five-phase system applied to the radial flux permanent magnet generator discussed above, the most qualified axial flux permanent magnet generator in [10] was modified. Modifications were made by increasing the number of stator cores, changing the position of the stator windings, and increasing the number of phase systems with the aim of upgrading the output power and power density. Besides the modification, optimization was also carried out to decrease the total harmonic distortion (THD) of the on-load voltage ($V_t$) of the generator by employing the Taguchi and Artificial Neural Network (ANN) methods. The content of this paper is divided into four sections. Section 1 discusses the introduction. Section 2 discusses the proposed five-phase axial flux permanent magnet generator (FP-AFPMG) and its optimization process. The results and discussion are reported in Section 3, and completed with the conclusion in Section 4.

## 2. Proposed FP-AFPMG and Optimization Process

### 2.1. Proposed FP-AFPMG

In order to upgrade the output power ($P_{out}$) of the three-phase AFPMG shown in Figure 1 for standalone wind turbine power generation in remote areas, some improvements were added without changing the number of the stator winding turns and the diameter of the conductor in each phase.

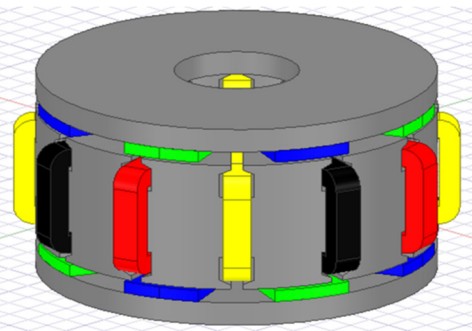

**Figure 1.** Original shape of the most qualified axial flux permanent magnet generator in [10].

The first improvement shown in Figure 2 is carried out by adding one stator core placed on the upper side and one stator core placed on the lower sides of the original

generator. The stator cores are added in order to upgrade the output power ($P_{out}$) and to limit the diameter of the generator. The number of stator slots and the dimension of the stator slot are equal with the original generator. The thickness of the stator yoke on the upper and lower side of the generator shown in Figure 2 is half the thickness of the stator yoke of the original generator.

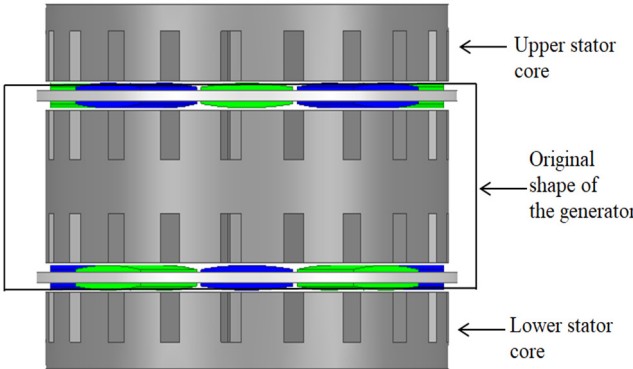

**Figure 2.** Modified shape of the original generator in Figure 1.

The second improvement is carried out by changing the winding type from core wound to tooth wound. Due to the addition of the stator core on the upper and lower sides of the original generator, the number of the flux lines flowing on the middle stator yoke is higher than the upper and lower stator yokes, as illustrated in Figure 3.

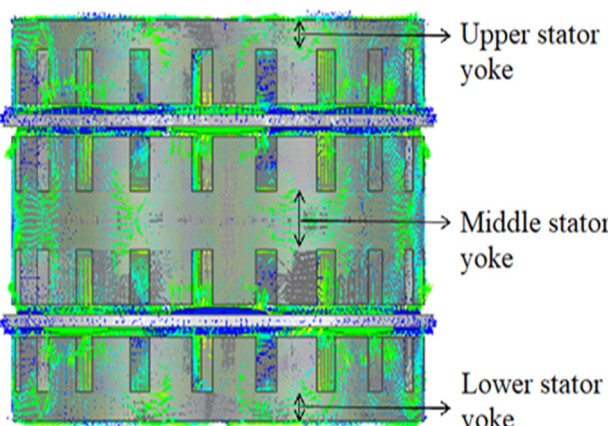

**Figure 3.** Flux lines in the magnetic circuit of the proposed generator.

When using the core wound winding type, the voltage ($E_f$) generated in each coil located in layer 2 is higher than the coil located in layer 1 and layer 3, as shown in Figure 4a. To generate equal $E_f$, the winding type needs to be changed from core wound to tooth wound, as shown in Figure 4b. By generating equal $E_f$ between each of the coils located in layer 1, layer 2, layer 3, and layer 4 of Figure 4b, the coil can be more easily connected in parallel or in series.

The third improvement is carried out by modifying the permanent magnet shape, as shown in Figure 5. Figure 5a,c is the permanent magnet shapes of the original generator. To simplify the manufacturing process of the permanent magnet in Figure 5a, its shape is modified to the permanent magnet shape shown in Figure 5b. Furthermore, the arc-shaped permanent magnet (PM) with an arc height of 2 mm, as shown in Figure 5d, is employed in order to make the flux density in the air gap more sinusoidally distributed. By improving the flux density waveform close to a sinusoidal waveform, the THD of the on-load voltage waveform ($V_t$) can be improved.

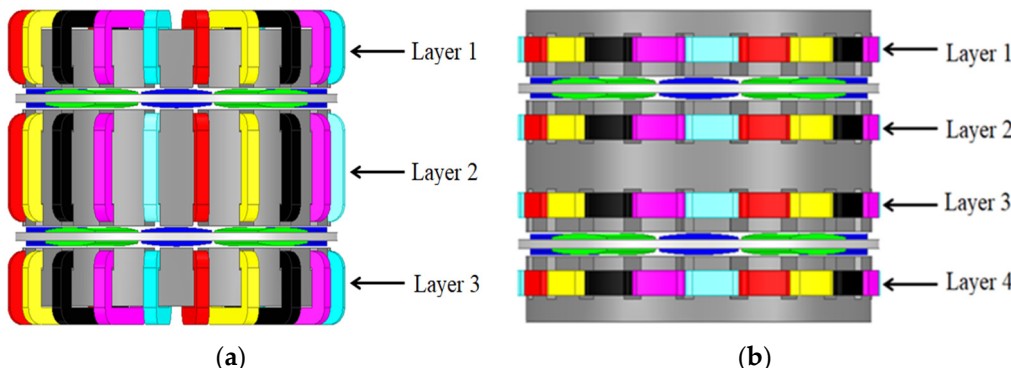

**Figure 4.** Winding type (**a**) core wound, (**b**) tooth wound.

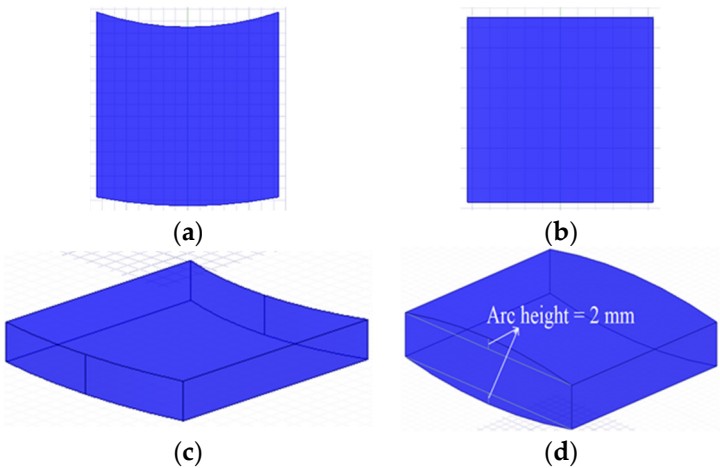

**Figure 5.** Modification of permanent magnet pole. (**a**,**c**) original shape of the permanent magnet pole, (**b**,**d**) modified shape of the permanent magnet pole.

Along with the rapid development of power electronic devices and efforts to improve generator performances, the last improvement is carried out by changing the phase system of the original generator into a five-phase system. As a five-phase system is categorized into a multiphase system, the power density ($P_{density}$) of the generator can be increased and the ripple torque produced by the generator can be decreased, as stated in Section 1. To restrict the flux density in the stator teeth at around 1.5 T to 2 T and in the stator yoke at around 1.1 T to 1.5 T, according to [11], the generator is firstly simulated in a magnetostatic solution type from the Ansys Maxwell software under a constant permanent magnet dimension and a constant air gap length. After the simulation is complete, the dimensions of the generator were tabulated in Table 1 and illustrated in a three-dimensional model in Figure 6.

**Table 1.** Dimensions of the proposed generator.

| No | Part | Dimension (mm) |
|---|---|---|
| 1 | Inner and outer diameters | 135 and 233 |
| 2 | Air gap length | 1 |
| 3 | Upper and lower sides stator yoke | 19 |
| 4 | Middle side stator yoke | 38 |
| 5 | Stator slot width | 12 |
| 6 | Stator slot height | 20 |
| 7 | Stator slot wedge height | 2 |
| 8 | Stator slot opening height | 2 |
| 9 | Stator slot opening width | 4.5 |
| 10 | Edge thickness of the permanent magnet | 8 |

**Table 1.** *Cont.*

| No | Part | Dimension (mm) |
|---|---|---|
| 11 | Arc height of the permanent magnet | 2 |
| 12 | Pole width to pole pitch ratio | 0.95 |
| 13 | Total axial length of the FP-AFPMG | 175 |

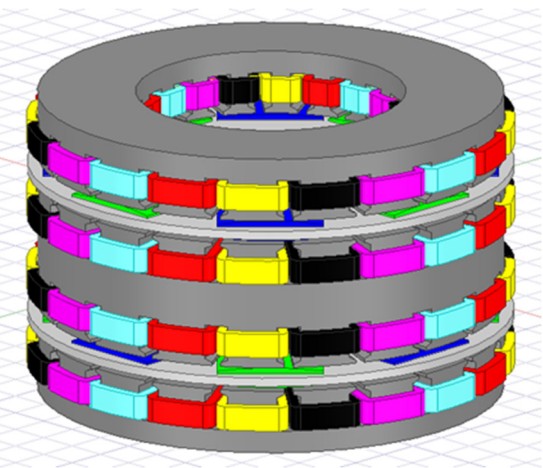

**Figure 6.** Modified shape of the axial flux permanent magnet generator in Figure 1.

In this study, the stator teeth of the generator were wound with a double-layer non-overlap stator winding type due to the several advantages possessed by this winding type as stated in [12]. The number of stator coils per phase required to implement the double-layer non-overlap winding type is counted by Equation (1) [13]. The machine periodicity and the number of spokes can be calculated by Equations (2) and (3). Equation (4) formulates the electric angle between two phasors in Figure 7a, and Equation (5) computes the angle of the sector in Figure 7a [14].

$$n_c = \frac{N_s}{m} \tag{1}$$

$$\delta = GCD\ (N_s,\ p) \tag{2}$$

$$Spoke = \frac{N_s}{\delta} \tag{3}$$

$$\alpha_e = p\ \frac{2\pi}{N_s} \tag{4}$$

$$\alpha_s = \frac{\pi}{m} \tag{5}$$

In Equations (1)–(5), $N_s$ is the stator slot number, $m$ is the phase number, $GCD$ is the greatest common divisor, and $p$ is the pole pairs. With the combination of twenty stator slots and eight PMs, the calculated $\delta$ obtained from Equation (2) is 4 and the calculated $\alpha_e$ obtained from Equation (4) is $2\pi/5$. Thus, the star of slots includes five spokes in which each spoke geometrically shifts by $\alpha = 2\pi/5$ and each spoke consists of four phasors, as shown in Figure 7a. In the case of $\alpha_e = \alpha$, the positive phasors of each phase (phase a+, b+, c+, d+, and e+) are located in one sector, so the opposite sectors (negative phasor of phase a-, b-, c-, d-, and e-) are vacant. The winding arrangement for each phase converted from the phasor diagram in Figure 7a is shown in Figure 7b. As the opposite sector of the positive phasor is vacant, the winding polarity of each phase will not be differently wound in each slot, as shown in Figure 7b.

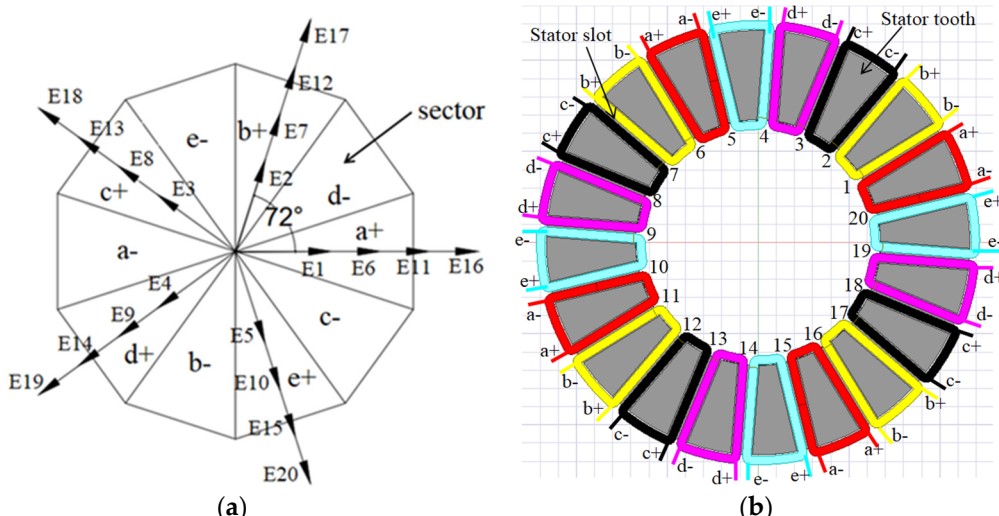

(a)            (b)

**Figure 7.** Five-phase stator winding configuration. (**a**) Star of slot, (**b**) armature winding arrangement in the stator slot.

### 2.2. Optimization of Rotor Poles

To reduce the THD of the AFPMG, [15] proposes the modification of the permanent magnet by adding an arc height facing the air gap. From their study, the THD of the generated voltage ($E_f$) can be decreased by 26.3% because of more sinusoidal flux density in the air gap when compared to the conventional model. Besides arc height, in this study, the pole width to pole pitch ratio ($\alpha_i$) and the slot opening width ($s_o$) depicted in Figure 8a,b, respectively, are also modified. Four different $\alpha_i$ of 0.8, 0.85, 0.9, 0.95 and four different $s_o$ of 4.5 mm, 7 mm, 9.5 mm, 12 mm are proposed in this study. The results of THD obtained from a transient solution type by connecting the generator with a resistive load are tabulated in Table 2.

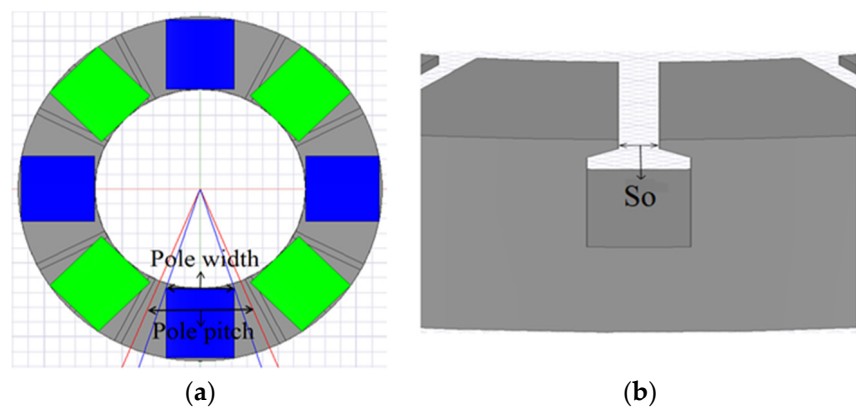

(a)            (b)

**Figure 8.** Proposed optimization of rotor pole. (**a**) Pole width to pole pitch ratio ($\alpha_i$), (**b**) stator slot opening width ($s_o$).

**Table 2.** Harmonic magnitude of the combination between $\alpha_i$ and $s_o$.

| $\alpha_i$ | $s_o$ | Harmonic Order | | | | | | THD (%) |
| --- | --- | --- | --- | --- | --- | --- | --- | --- |
| | | **1** | **3** | **7** | **9** | **11** | **13** | |
| | 4.5 | 39.3687 | 8.308 | 0.1955 | 0.4424 | 0.0317 | 0.0872 | 21.1401 |
| 0.8 | 7 | 38.6705 | 8.7878 | 0.2238 | 0.4597 | 0.0187 | 0.0846 | 22.76435 |
| | 9.5 | 37.6329 | 9.1347 | 0.2907 | 0.4261 | 0.045 | 0.0316 | 24.31228 |
| | 12 | 36.3404 | 9.3068 | 0.42 | 0.3252 | 0.0554 | 0.0071 | 25.6522 |

**Table 2.** *Cont.*

| $\alpha_i$ | $s_o$ | Harmonic Order | | | | | | THD (%) |
|---|---|---|---|---|---|---|---|---|
| | | **1** | **3** | **7** | **9** | **11** | **13** | |
| 0.85 | 4.5 | 40.2488 | 5.2655 | 0.4533 | 0.371 | 0.137 | 0.1078 | 13.1702 |
| | 7 | 39.63 | 5.6814 | 0.3924 | 0.4114 | 0.0601 | 0.0779 | 14.40985 |
| | 9.5 | 38.5243 | 6.0454 | 0.2876 | 0.418 | 0.0268 | 0.0397 | 15.7481 |
| | 12 | 37.1757 | 6.2828 | 0.1638 | 0.3741 | 0.0435 | 0.0041 | 16.93636 |
| 0.9 | 4.5 | 41.0784 | 2.1977 | 0.9258 | 0.3008 | 0.1536 | 0.0681 | 5.865619 |
| | 7 | 40.3727 | 2.5908 | 0.7778 | 0.2671 | 0.0833 | 0.0373 | 6.736538 |
| | 9.5 | 39.277 | 2.9472 | 0.5874 | 0.318 | 0.0565 | 0.0272 | 7.695586 |
| | 12 | 37.8461 | 3.2328 | 0.2642 | 0.313 | 0.0308 | 0.0063 | 8.610653 |
| 0.95 | 4.5 | 41.7142 | 0.8556 | 1.1187 | 0.141 | 0.1477 | 0.028 | 3.412228 |
| | 7 | 40.9652 | 0.5256 | 1.0304 | 0.0947 | 0.0874 | 0.0217 | 2.841605 |
| | 9.5 | 39.8307 | 0.2552 | 0.8146 | 0.178 | 0.0694 | 0.0104 | 2.196344 |
| | 12 | 38.3179 | 0.2925 | 0.469 | 0.2131 | 0.022 | 0.0071 | 1.547171 |

From Table 2, the 5th harmonic is eliminated because the generator is connected in a star connection, and the 3rd harmonic is more dominant compared to the 7th, 9th, 11th, and 13th. Equation (6) is formulated to calculate the THD.

$$\text{THD} = \frac{\sqrt{V_2^2 + V_3^2 + V_4^2 + \cdots}}{V_1^2} \tag{6}$$

In Equation (6), $V_n$ is the nth harmonic of the terminal voltage ($V_t$) and $V_1$ is the fundamental harmonic of $V_t$. From Table 2, because the combination of $\alpha_i$ = 0.95 and $s_o$ = 12 mm gives the smallest THD of 1.55%, this combination is chosen to be the optimal rotor pole combination in this case.

### 3. Taguchi and Artificial Neural Network (ANN) Methods

The Taguchi method was formulated by Sir Ronald Fisher in the 1920s [16]. The method works by using the orthogonal array to analyze the information from the control factors and the levels to gain useful static information in the fewest experiments [17]. Knowing the number of the control factors and the levels, the correct orthogonal array can be chosen from the references. The generator with optimal rotor combination obtained from the process in Section 2.2, shown in Figure 9, is optimized again by the Taguchi method to acquire the lowest THD. As the lower and upper parts of the region marked by red are the same, the generator can be split in a symmetrical position to simplify its geometry, as shown in Figure 10.

From Figure 10, the areas chosen as the control factors to improve the THD are marked with A, B, and C. The area marked with A indicates the air gap length, the area marked with B indicates the offset angle of the stator teeth on the upper side to the stator teeth on the lower side, and the area marked with C indicates the stator yoke height. The level of each control factor is tabulated in Table 3. The air gap lengths of 1 mm, 1.5 mm, 2 mm, and 2.5 mm are the levels of control factor (A). The offset angles of 0 degree, 1.5 degrees, 3 degrees, and 4.5 degrees in mechanical degree are the levels of control factor (B). The stator yoke heights of 19 mm, 17 mm, 15 mm, and 13 mm are the levels of control factor C. As there are four levels for each control factor, the $L_{16}(4^3)$ sequence of the Taguchi orthogonal array is selected and the proper combinations of each level of the control factors are tabulated in Table 4. In the optimization process, the mean and variance of the experimental result obtained from Table 4, in this case, are combined into a single performance measure called the signal to noise (S/N) ratio.

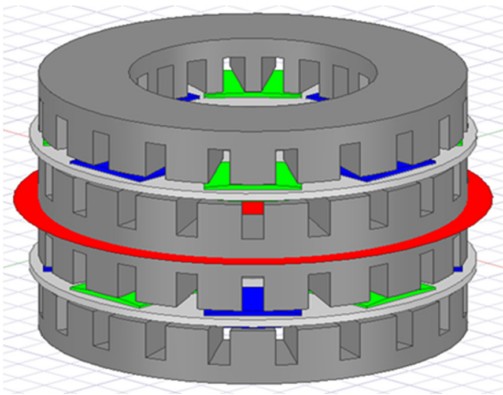

**Figure 9.** The generator to be optimized by Taguchi method.

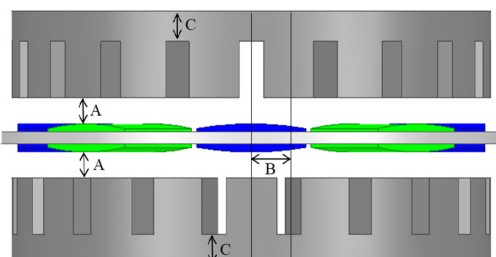

**Figure 10.** Symmetrical position of the generator.

**Table 3.** Control factors and levels of FP-AFPMG.

| Control Factor | | Levels of Control Factor | | | |
|---|---|---|---|---|---|
| | | **1** | **2** | **3** | **4** |
| Air gap length (mm) | A | 1 | 1.5 | 2 | 2.5 |
| Offset angle of stator tooth (mechanical degree) | B | 0 | 1.5 | 3 | 4.5 |
| Stator yoke height (mm) | C | 19 | 17 | 15 | 13 |

**Table 4.** Taguchi $L_{16}$ sequence.

| No | Control Factor | | |
|---|---|---|---|
| | **A** | **B** | **C** |
| 1 | 1 (1) | 0 (1) | 19 (1) |
| 2 | 1 (1) | 1.5 (2) | 17 (2) |
| 3 | 1 (1) | 3 (3) | 15 (3) |
| 4 | 1 (1) | 4.5 (4) | 13 (4) |
| 5 | 1.5 (2) | 0 (1) | 17 (2) |
| 6 | 1.5 (2) | 1.5 (2) | 19 (1) |
| 7 | 1.5 (2) | 3 (3) | 13 (4) |
| 8 | 1.5 (2) | 4.5 (4) | 15 (3) |
| 9 | 2 (3) | 0 (1) | 15 (3) |
| 10 | 2 (3) | 1.5 (2) | 13 (4) |
| 11 | 2 (3) | 3 (3) | 19 (1) |
| 12 | 2 (3) | 4.5 (4) | 17 (2) |
| 13 | 2.5 (4) | 0 (1) | 13 (4) |
| 14 | 2.5 (4) | 1.5 (2) | 15 (3) |
| 15 | 2.5 (4) | 3 (3) | 17 (2) |
| 16 | 2.5 (4) | 4.5 (4) | 19 (1) |

In obtaining the optimal quality of the desired parameter, there are three types of the signal to noise ratio (S/N) function that can be applied to solve the issue discussed in this study. Equations (7)–(9) give the condition of the smaller the better (STB), the larger the better (LTB), and the nominal the best (NTB) [17,18].

$$\frac{S}{N} = -10 \log \left[ \frac{1}{n} \sum_{i=1}^{n} \left( y_i^2 \right) \right] \tag{7}$$

$$\frac{S}{N} = -10 \log \left[ \sum_{i=1}^{n} \left( \frac{1}{y_i^2} \right) \right] \tag{8}$$

$$\frac{S}{N} = -10 \log \left[ \frac{y^2}{s^2} \right] \tag{9}$$

In Equations (7)–(9) $n$, $y_i$, $y$, and $s^2$ are the number of experiments, $i$th observation values, average of values, and variances of data respectively. In this study, Equation (7) is employed to achieve THD below 1%.

The Artificial Neural Network (ANN) is an intelligent computing system that was developed by McCulloch and Pitts [19]. The principle of the ANN works by imitating the capability of the human brain to learn from observations and to generalize by abstractions [20]. The ANN network consists of three layers, namely input layer, output layer, and hidden layer, where the neurons are accommodated [21]. The data obtained from the observation are collected in the input layer, useful information that is used to solve the non-linear problems is collected in the output layer, and the hidden layers are there to link the input layer and output layer and to process the data from the input layer [22]. Due to its successful ability to solve non-linear problems, the ANN is applied in many different applications, as described in [23]. Figure 11 shows the illustration of the ANN network, in this case, obtained based on reference [24], which is used to compare the result of the Taguchi optimization.

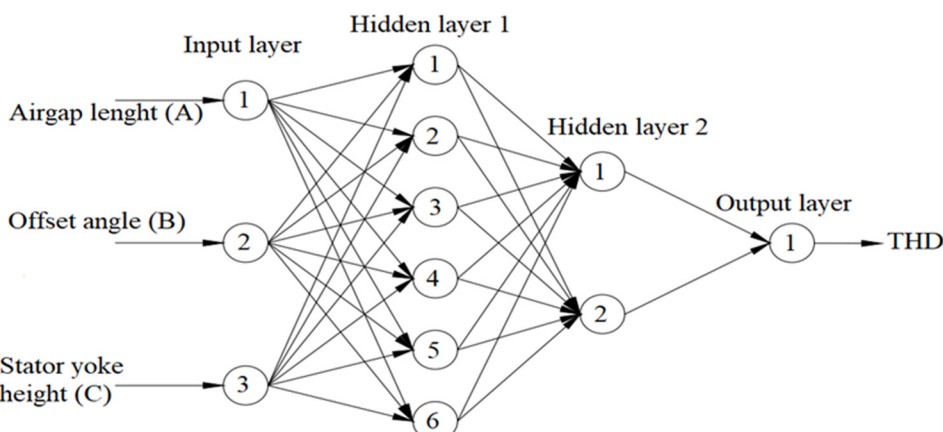

**Figure 11.** Structure of the Artificial Neural Network (ANN) used in this case.

From Figure 11, the input layer consists of three neurons in which neuron number 1 is for the air gap length (A), neuron number 2 is for the offset angle (B), and neuron number 3 is for the stator yoke height (C). Hidden layer 1 consists of six neurons and hidden layer 2 consists of two neurons. The output layer consists of one neuron to collect the useful information from the hidden layer to predict THD. The ANN tool of Matlab R2020 is used to implement the structure of the ANN network in Figure 11. The network type of a feed-forward backpropagation with a trained function of Levenberg–Marquardt (LM) and a logsig transfer function for two hidden layers are used to predict THD. Input and target data obtained from the Taguchi orthogonal array are automatically divided by the ANN tool into three groups: training data (75%), testing data (12.5%), and validation

data (12.5%). The quality of the ANN network is evaluated by the percentage of prediction error formulated by Equation (10) [22].

$$\% \ error = \left| \frac{Experimental \ data - ANN \ data}{Experimental \ data} \right| * 100 \ \% \tag{10}$$

In the case of (10), experimental data are data obtained from the Taguchi combination and ANN data are obtained from THD prediction of the ANN network.

## 4. Results and Discussion

In this study, the most qualified axial flux permanent magnet generator (AFPMG) in [10] was upgraded, to be employed for standalone residential wind turbine power generation. Some improvements were carried out on its geometry, such as adding two stator cores placed on the upper and lower sides of the original generator, modifying the permanent magnet shape, changing the winding type from core wound to tooth wound, and substituting a three-phase system with a five-phase system, as explained in Section 2.1. By restricting the flux density in the stator teeth and in the stator yoke, the dimensions of the five-phase AFPMG (FP-AFPMG) were gained with the help of Ansys Maxwell software in the magnetostatic solution type. Then, the rotor pole was optimized to improve THD by varying $S_o$ and $\alpha_i$. With the combination of $S_o$ of 12 mm and $\alpha_i$ of 0.95, the optimal rotor combination was achieved with a THD of 1.55%, as described in Section 2.2. With the purpose of generating the lowest THD, the proposed generator with optimal rotor combination was optimized again by the Taguchi method. By applying the Taguchi method in this case, a lot of time can be saved, especially for a simulation with a three-dimensional model if compared to the conventional experimental method. As explained in Section 3, three control factors, including the air gap length, the offset angle of the stator teeth on the upper side to the stator teeth on the lower side, and the stator yoke height were selected as independent control factors with four levels in each control factor. The measured THD and the calculated S/N ratio generated by the $L_{16}$ Taguchi table are shown in Table 5.

**Table 5.** Results of $L_{16}$ Taguchi table.

| No | A | B | C | THD (%) | S/N |
|----|------|---------|---------|---------|-------|
| 1 | 1 (1) | 0 (1) | 19 (1) | 1.55 | −3.79 |
| 2 | 1 (1) | 1.5 (2) | 17 (2) | 1.28 | −2.12 |
| 3 | 1 (1) | 3 (3) | 15 (3) | 1.10 | −0.83 |
| 4 | 1 (1) | 4.5 (4) | 13 (4) | 0.66 | 3.63 |
| 5 | 1.5 (2) | 0 (1) | 17 (2) | 1.63 | −4.25 |
| 6 | 1.5 (2) | 1.5 (2) | 19 (1) | 1.44 | −3.18 |
| 7 | 1.5 (2) | 3 (3) | 13 (4) | 1.30 | −2.28 |
| 8 | 1.5 (2) | 4.5 (4) | 15 (3) | 0.87 | 1.22 |
| 9 | 2 (3) | 0 (1) | 15 (3) | 1.58 | −3.95 |
| 10 | 2 (3) | 1.5 (2) | 13 (4) | 1.51 | −3.60 |
| 11 | 2 (3) | 3 (3) | 19 (1) | 1.26 | −2.01 |
| 12 | 2 (3) | 4.5 (4) | 17 (2) | 0.98 | 0.21 |
| 13 | 2.5 (4) | 0 (1) | 13 (4) | 1.58 | −3.99 |
| 14 | 2.5 (4) | 1.5 (2) | 15 (3) | 1.49 | −3.48 |
| 15 | 2.5 (4) | 3 (3) | 17 (2) | 1.26 | −1.98 |
| 16 | 2.5 (4) | 4.5 (4) | 19 (1) | 1.02 | −0.14 |

The THD in Table 5 is measured when the proposed generator is simulated in the transient solution type from the Ansys Maxwell software in which the generator is connected to the resistive load. The S/N ratio is obtained based on the function of the smallest the better (STB). The most effective control factor that influences THD is obtained by the S/N ratio respond curve shown in Figure 12. From Figure 12, based on the function of the smaller the better, the most effective control factors are A4, B1, and C1. A4 means the level of air gap length at level number 4, B1 means the level of the stator teeth on the upper

side to the stator teeth on the lower side at level number 1, and C1 means the level of the stator yoke height at level number 1. From Table 5, the air gap length at level number 4 is 2.5 mm, the stator teeth on the upper side to the stator teeth on the lower side at level number 1 is 0 degree, and the stator yoke height at level number 1 is 19 mm. The simulation is then conducted based on the most effective control factor to determine THD. After THD is obtained from the most effective control factor, its magnitude is converted to the smaller the better S/N function with a magnitude of −4. According to the references in [25–27], the highest S/N ratio will result in a better performance irrespective of its characteristics. After all the S/N ratio magnitudes are compared for all the simulations, including the magnitude of the most effective control factor, the highest S/N ratio is obtained from the combination of A1, B4, and C3 with the magnitude of 3.63. Thus, the combinations of the air gap of 1 mm, the offset angle of the stator teeth on the upper side to the stator teeth on the lower side of 4.5 mechanical degrees, and the stator yoke height of 13 mm result in a THD of 0.66%.

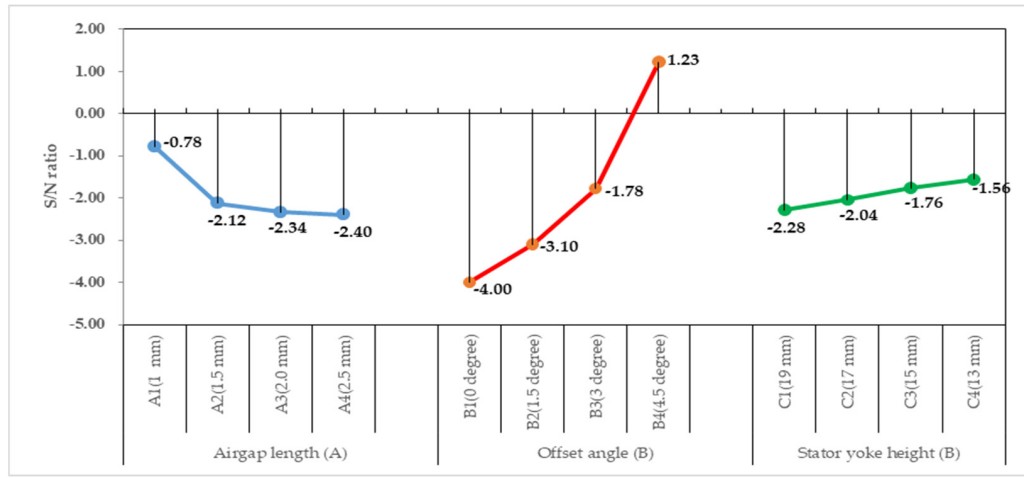

**Figure 12.** S/N ratio response curve of THD.

To compare the optimal result of the THD that is obtained from the Taguchi method, the Artificial Neural Network (ANN) built from Matlab R2020b was utilized in this study. To describe the ANN network in Figure 11, the nntool function is called in the Matlab command window. The user should firstly categorize the data as input and target data and then they should create an ANN network that describes the network in Figure 11. In this study, the sixteen combinations in Table 4 were categorized as input data, and the target data are the S/N ratio in Table 5. By using a feed-forward backpropagation network trained with Levenberg–Marquardt (LM) and a logsig transfer function for the two hidden layers, the results of the network can be illustrated in Figure 13. By the nntool function, the network automatically splits the input and target data into 75% as training data, 12.5% as validation data, and 12.5% as testing data. After the network is trained with several repetitions, the predicted values of THD are found to be very close to the simulation results. This is proven by the regression analysis in Figure 14, in which the magnitude of the training, validation, test, and all data is close to 1.

Table 6 shows the comparison of the S/N ratio that resulted from the Taguchi and ANN methods. All predicted values of the S/N ratio resulting from the network shown in Figure 13 are almost the same as the S/N ratio resulting from the Taguchi method except for experiments number 8 and number 9. The predicted S/N ratio in experiment number 8 and in experiment number 9 are 1.62 and −4.09, respectively. There is a deviation of 0.4 in experiment number 8 and a deviation of 0.14 in experiment number 9 when compared to the Taguchi method. Based on Equation (10), the percent error of experiment number 8 is 32.87% and this is 3.54% for experiment number 9. If the mean error is calculated for all the experiments (16 experiments), a mean error of 2.28% is calculated, which is less than

3%. This indicates that the Taguchi results and predicted results of the ANN network have an excellent relationship, as illustrated in Figure 15. Hence, the results also show that the network architecture of 3-6-2-1 for the input neuron of the air gap length, the offset angle of the stator teeth on the upper side to the stator teeth on the lower side, and the stator yoke height in this case is able to predict THD with less error percentage.

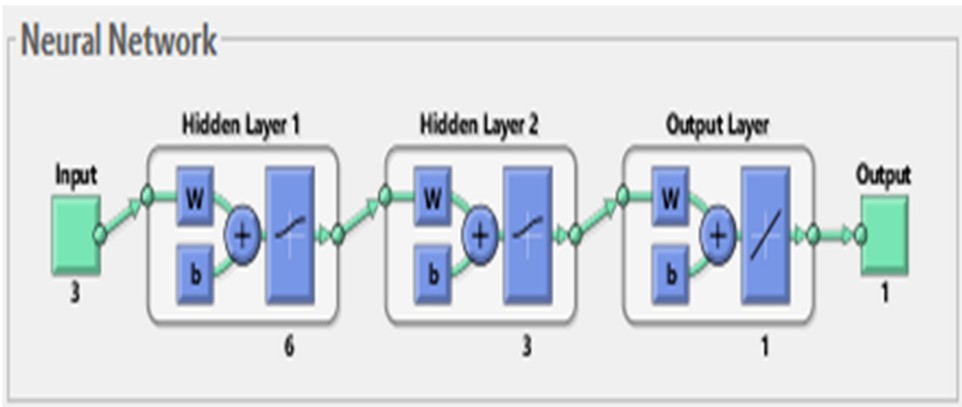

**Figure 13.** Design of the ANN network in Matlab R2020b toolbox.

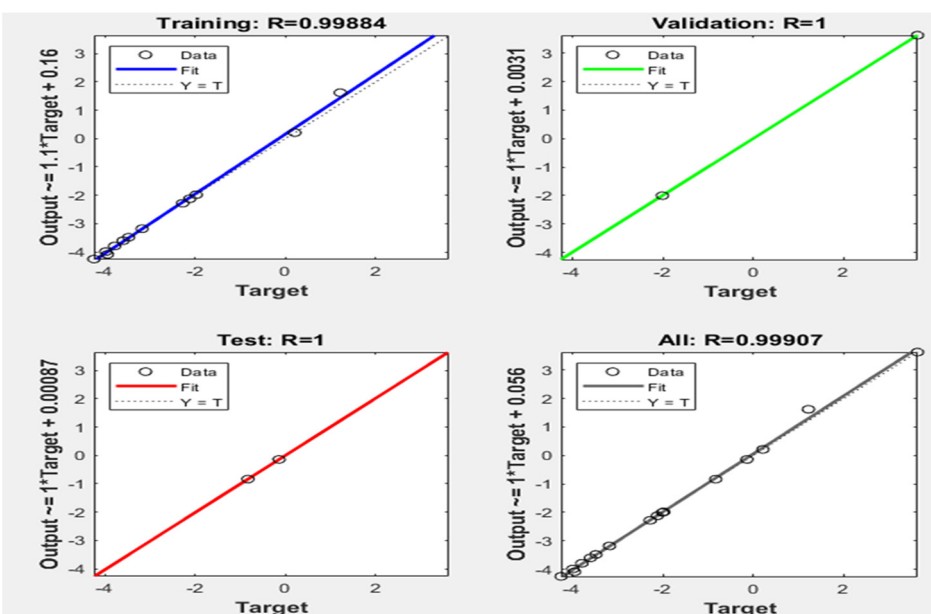

**Figure 14.** Regression analysis of the THD.

**Table 6.** Comparison of S/N ratio between Taguchi and ANN.

| No | A | B | C | Taguchi S/N | ANN S/N | Error (%) |
|---|---|---|---|---|---|---|
| 1 | 1 | 0 | 19 | −3.79 | −3.79 | 0.00 |
| 2 | 1 | 1.5 | 17 | −2.12 | −2.12 | 0.00 |
| 3 | 1 | 3 | 15 | −0.83 | −0.83 | 0.00 |
| 4 | 1 | 4.5 | 13 | 3.63 | 3.63 | 0.00 |
| 5 | 1.5 | 0 | 17 | −4.25 | −4.25 | 0.00 |
| 6 | 1.5 | 1.5 | 19 | −3.18 | −3.18 | 0.00 |
| 7 | 1.5 | 3 | 13 | −2.28 | −2.28 | 0.00 |
| 8 | 1.5 | 4.5 | 15 | 1.22 | 1.62 | 32.87 |
| 9 | 2 | 0 | 15 | −3.95 | −4.09 | 3.54 |
| 10 | 2 | 1.5 | 13 | −3.60 | −3.60 | 0.00 |

**Table 6.** *Cont.*

| No | A | B | C | Taguchi S/N | ANN S/N | Error (%) |
|---|---|---|---|---|---|---|
| 11 | 2 | 3 | 19 | −2.01 | −2.01 | 0.00 |
| 12 | 2 | 4.5 | 17 | 0.21 | 0.21 | 0.00 |
| 13 | 2.5 | 0 | 13 | −3.99 | −3.99 | 0.00 |
| 14 | 2.5 | 1.5 | 15 | −3.48 | −3.48 | 0.00 |
| 15 | 2.5 | 3 | 17 | −1.98 | −1.98 | 0.00 |
| 16 | 2.5 | 4.5 | 19 | −0.14 | −0.14 | 0.00 |
| | | | | | |*Error sum*| 36.41 |

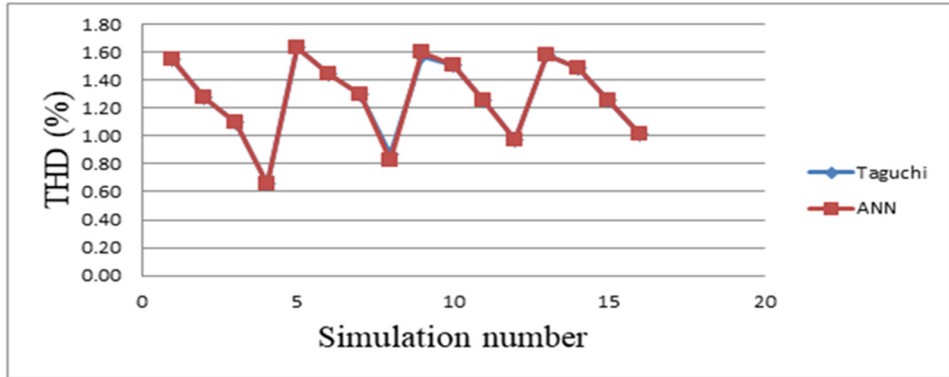

**Figure 15.** Comparison of Taguchi and ANN.

The performances of the proposed generator measured by the Ansys Maxwell software are shown below.

- Distribution of magnetic flux density

The distribution of the magnetic flux density is illustrated in Figure 16. Figure 16a shows its distribution in the symmetrical position of the proposed generator. To clearly see its distribution from Figure 16a, the stator cores located on the lower and upper sides are separated as shown in Figure 16b,c. To avoid the phenomena of demagnetizing the permanent magnet and to avoid increasing loss and temperature in the stator core, the maximum flux density in the stator core is firstly checked so that its magnitude does not exceed the permissible limit. In the stator teeth of Figure 16b,c, the maximum flux densities are 1.55 T and 1.6 T, respectively. The flux densities of 1.27 T and 1.28 T are the maximum flux density in the stator yoke of Figure 16b,c, respectively. If compared to reference [11], the maximum flux density on both the stator cores meets the requirement stated in reference [11]. This signifies that the phenomena of demagnetizing the permanent magnet can be avoided and the increasing loss and temperature in the stator core can be minimized in the proposed generator.

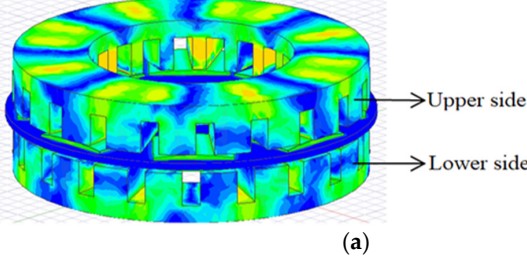

(**a**)

**Figure 16.** *Cont.*

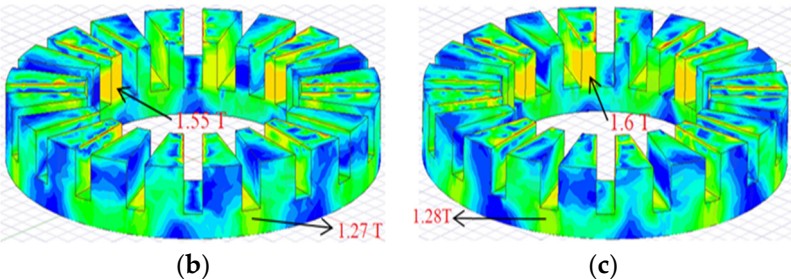

**Figure 16.** Flux density distribution of the proposed generator. (**a**) Symmetrical position, (**b**) lower side of the stator core, (**c**) upper side of the stator core.

- Voltage waveform

The voltage waveform of the original generator and the proposed generator are shown in Figure 17a,b, respectively. Their measured harmonic orders are represented in Figure 17c. If compared to the original generator, the voltage waveform of the proposed generator is almost a pure sinusoidal waveform. In terms of the harmonic order from Figure 17c, as both generators are connected in a star connection, the 3rd harmonic is zero in the original generator and the 5th harmonic is zero in the proposed generator. The original generator generates the 5th harmonic of 2.391 volts and the proposed generator generates the 3rd harmonic of 0.2267 volts. The 7th, 9th, 11th, and 13th harmonic orders of the proposed generator all decrease with an average percentage of 71.08% when compared to the original generator. The calculated THD obtained from Equation (6) of the original generator is 8.53% and this is 0.65% for the proposed generator. From those results, a good improvement is achieved in term of THD for the proposed generator.

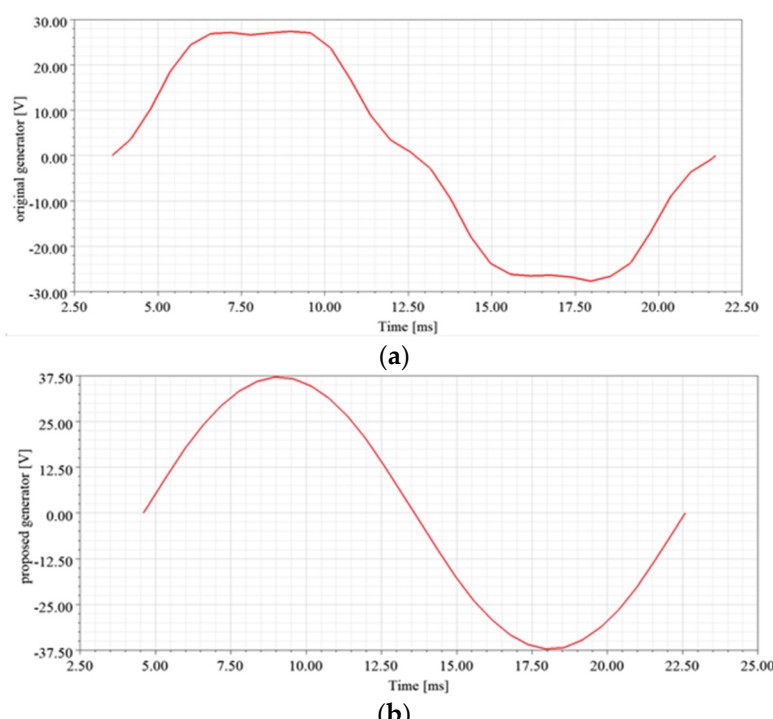

**Figure 17.** *Cont.*

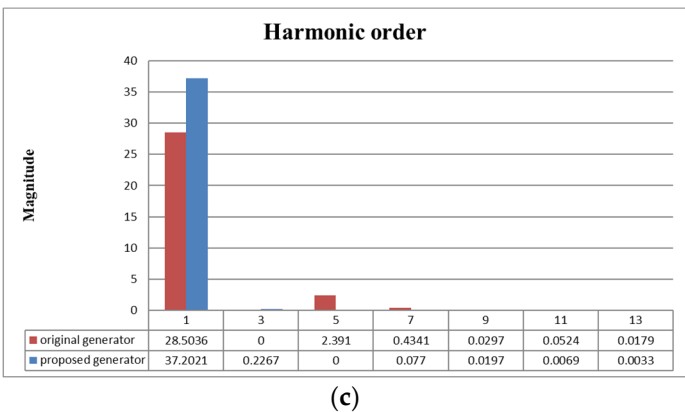

(c)

**Figure 17.** Voltage waveform and harmonic order; (**a**) original generator, (**b**) proposed generator, (**c**) harmonic order of original and proposed generators.

- Moving torque

The moving torque of the original generator and the moving torque of the proposed generator are depicted in Figure 18. The green line indicates the moving torque of the original generator and red lines indicate the moving torque of the proposed generator. Moving torque is produced because of the interaction between a magnetic field from the rotor circuit and another magnetic field from the stator circuit when the load current flows in the circuit. Due to the lowest THD of the proposed generator, the ripples in the moving torque are lower than the ripples in the original generator. The percentage of the ripple torque can be calculated by comparing the difference between the maximum and minimum torques to the average torque. From Figure 18, the calculated ripple torque of the proposed generator is 2.42% and this is 45.82% for the original generator. From the results, a good improvement of 94.72% in the torque ripple is obtained by the proposed generator. With the lowest torque ripple, the generator will rotate in a smooth performance, which can minimize the bearing fatigue.

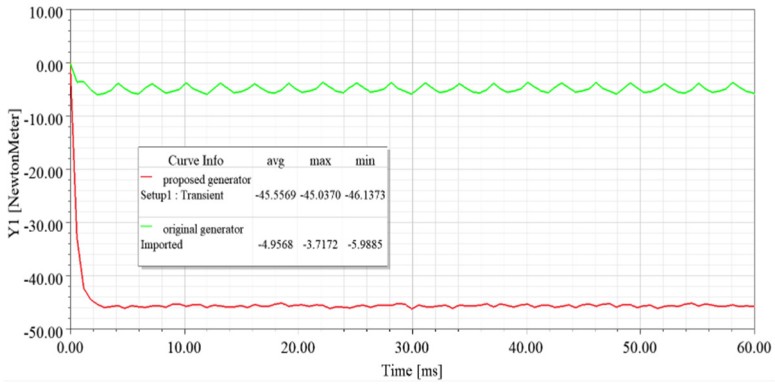

**Figure 18.** Moving torque of original generator and proposed generator.

- Cogging torque

Cogging torque is also known as detent torque, depicted in Figure 19. It appears in the permanent magnet machines as an interaction between the permanent magnets on the rotor core and the slots in the stator core. It happens when the stator windings are not energized. Cogging torque is an undesirable effect on permanent magnet generators as it can affect the self-starting capability of the generator, generate mechanical vibration, and produce acoustic noise. For small wind turbine generator applications, the cogging torque generated from the generator is expected to be low enough so that the aerodynamic power of the turbine in the start-up position can overcome it. The percentage of the cogging torque

factor is defined as the ratio of the peak-to-peak cogging torque to the average torque. From Figure 19, the calculated cogging torque factor for the original generator is 55.15% and this is 1.75% for the proposed generator. The results show a good improvement of 96.83% in term of the cogging torque factor, which will make the generator easier to rotate in the start-up position.

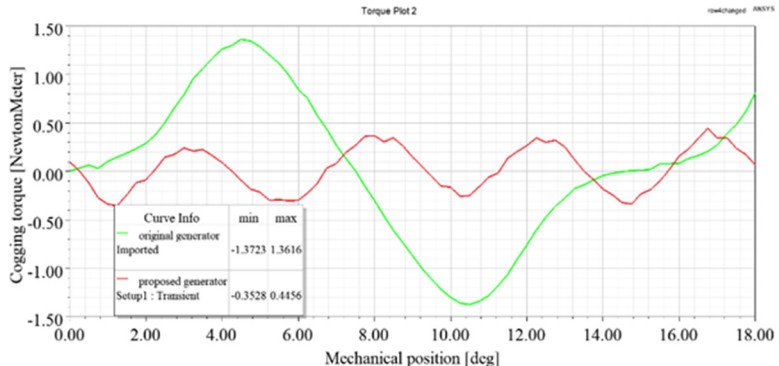

**Figure 19.** Cogging torque of original generator and proposed generator.

- Output power

The output power ($P_{out}$) of the original and proposed generators is shown in Figure 20. It is calculated as a product of the on-load terminal voltage ($V_t$), load current ($I_L$), power factor (cos $\varphi$), and the number of the system phase ($m_1$). In this study, as the generator was simulated to be connected to a resistive load so that the phase difference between $V_t$ and $I_L$ is zero, this results in cos $\varphi$ equal to 1. From Figure 20, $P_{out}$ of the original generator is 413, 91 watts, and is increased to 3.77 kW due to the effect of increasing $m_1$ and adding the stator core. Due to the addition of two stator cores, an increase in the number of stator slots from 12 slots to 20 slots, and an increase in the volume of the permanent magnet to maintain the flux density in the stator teeth and stator yoke of around 1.5–2 T and 1.1–1.5 T, respectively, the axial length and diameter of the original generator are increased, which results in an increase in volume. Power density ($P_{density}$) is defined as a measure of $P_{out}$ per volume of the generator. The calculated $P_{density}$ for the original generator is 212.69 kW/m$^3$ and this is 457.51 kW/m$^3$ for the proposed generator. From the results, it is shown that the purpose of upgrading $P_{out}$ of the original generator was fulfilled to support the independence of the electricity in remote areas where the electricity grid is not available.

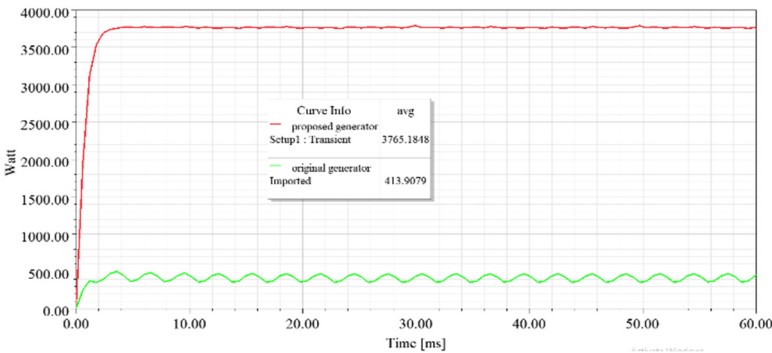

**Figure 20.** Output power of original generator and proposed generator.

- Efficiency

When the generator is connected to the load, load current will flow in the circuit. Some losses will be produced by the generator, which stops the efficiency of the generator from reaching 100%. The higher the losses generated by the generator, the lower the efficiency of

the generator, and vice versa. The efficiency can be calculated by comparing the maximum power delivered to the load (output power) to the total watts produced by the generator (input power). For an AFPMG in which the stator core is made of an iron core, some losses will be produced, including the stator winding losses, permanent magnet losses, rotor core losses, stator core losses, and friction and windage losses. In this study, the stator winding losses were calculated manually based on the data obtained from the simulation. Stator losses, rotor losses, and permanent magnet losses are calculated by the help of Ansys Maxwell software to acquire more accurate results. The friction and windage losses are calculated manually based on the equation explained in [10]. Table 7 shows the generated losses for the original generator and the proposed generator. From Table 7, it is clearly seen that when increasing $P_{out}$, the total losses produced by the generator also increase. The calculated efficiency of the proposed generator is 92.28%, in which there is an increasing magnitude of 0.30% of the original generator.

**Table 7.** Some losses produced by the original and proposed generators.

| No | Parameter | Original Generator (Watt) | Proposed Generator (Watt) |
|---|---|---|---|
| 1 | Stator winding losses | 20.84 | 200.73 |
| 2 | Stator core losses | 10.53 | 77.70 |
| 3 | Rotor core and permanent magnet | 0.1106 | 26.38 |
| 4 | Friction and windage losses | 4.4990 | 10.11 |
| 5 | Output power | 413.91 | 3765.18 |
| 6 | Input power | 449.89 | 4080.10 |

- Rectified voltage

In this study, it was attempted to connect the original generator and the proposed generator to a rectifier system, as shown in Figure 21. Figure 21a shows a schematic diagram of the co-simulation of the original generator connected to a full-wave three-phase rectifier and Figure 21b shows a schematic diagram of the co-simulation of the proposed generator connected to an uncontrolled five-phase rectifier.

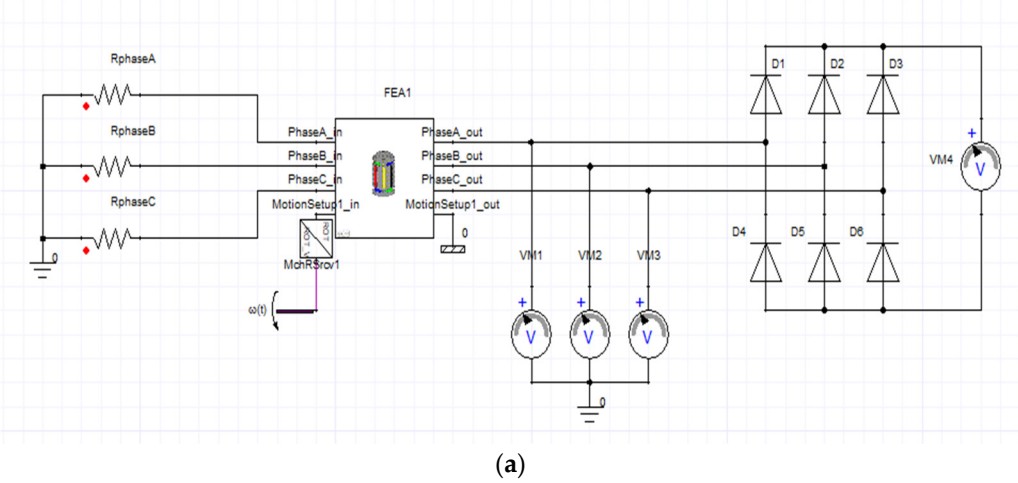

**(a)**

**Figure 21.** *Cont.*

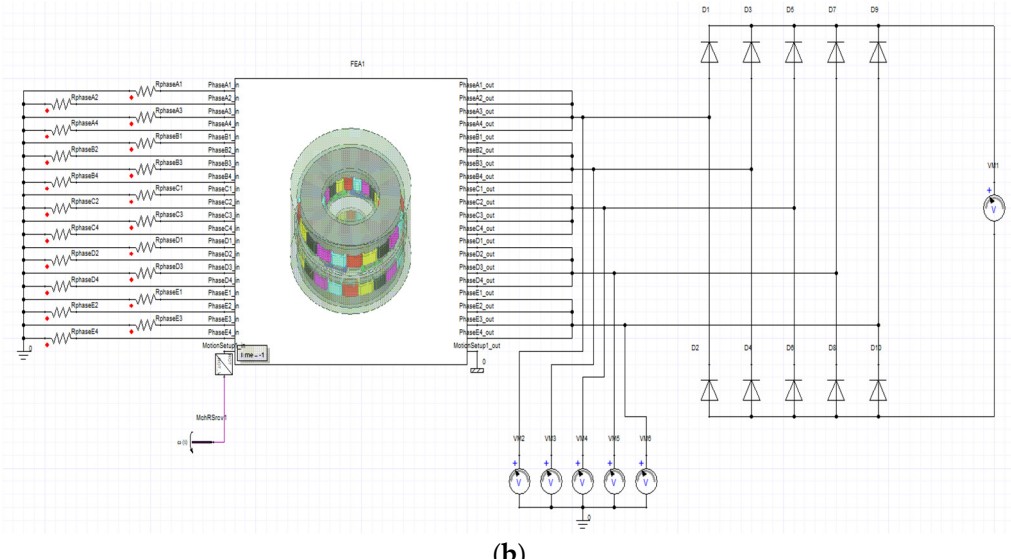

**(b)**

**Figure 21.** Schematic diagram of the co-simulation; (**a**) original generator, (**b**) proposed generator.

The co-simulation is carried out under a no-load condition to compare the magnitude of the no-load rectified voltage and its ripple resulting from the two systems. The rectified voltage of the original and proposed generators is shown in Figure 22. Figure 22a shows the input voltage and rectified voltage of the three-phase full-wave rectifier. The input voltage and rectified voltage of the five-phase uncontrolled rectifier is shown in Figure 22b. To prove the correctness of the co-simulation, the average magnitude of the rectified voltage for the two systems is compared with the theoretical equation. The average rectified voltage of the three-phase system is calculated by 1.65 $V_m$, and for the five-phase system this is calculated by 1.902 $V_m$ [28]. For the three-phase system, the measured magnitude is 50.17 V and the calculated magnitude is 48.09 V, and for the five-phase system, the measured magnitude is 72.88 V and the calculated magnitude is 75.45 V. The results show a slight deviation between the measured and calculated magnitudes for the two systems, which indicates that the co-simulation carried out for the two systems is correct. From Figure 22, it is shown that the rectified voltage of the five-phase system is smoother than the three-phase system. The calculated ripple of the three-phase rectifier is 32.34% and this is 3.63% for the five-phase rectifier. It shows that an improvement of 88.78% in the ripple of the rectified voltage is achieved by using the five-phase system, which reduces the size of the capacitor to smooth the rectified voltage.

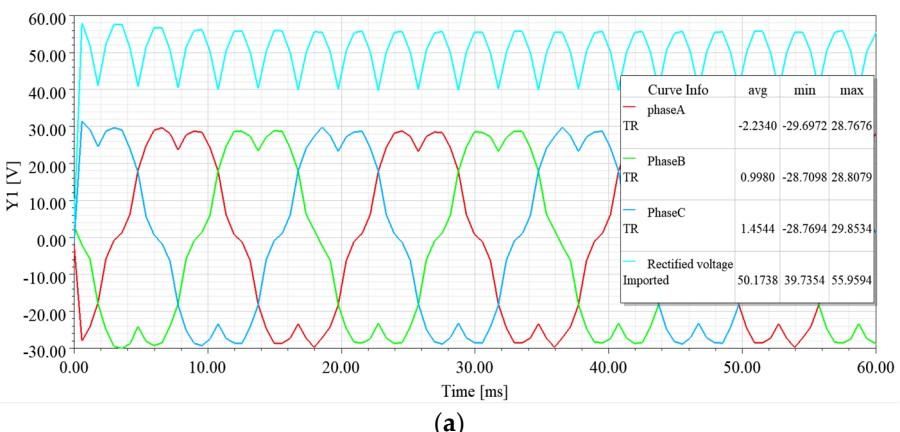

**(a)**

**Figure 22.** *Cont.*

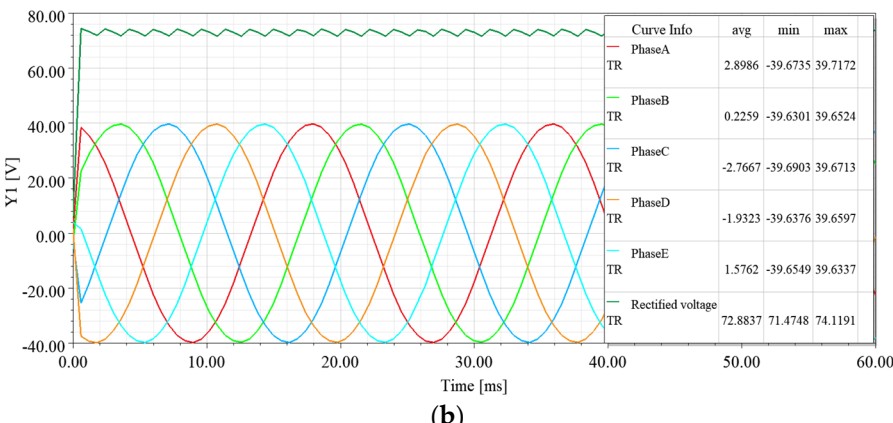

(**b**)

**Figure 22.** Input voltage and rectified voltage; (**a**) three-phase full-wave rectifier, (**b**) five-phase uncontrolled rectifier.

From the results and discussion above, there are several advantages of the proposed generator that can be achieved when compared to the original generator.

(1)    Low total harmonic distortion;
(2)    Low ripple torque;
(3)    Low cogging torque;
(4)    High output power;
(5)    High efficiency;
(6)    Low ripple of the rectified voltage.

From a mechanical point of view, the realization of the proposed generator is shown in Figure 23, and Figure 24 illustrates the exploded geometry of the proposed generator. The geometry of the proposed generator can be equipped with a cover, as shown in Figure 23a, or uncovered, as shown in Figure 23b. Uncovered geometry can provide easy heat transfer from the stator winding due to direct contact with the ambient temperature but this geometry is at risk from water and corrosion. Covered geometry is safe against water and corrosion but rather difficult in heat transfer from the stator windings because it is blocked by the cover.

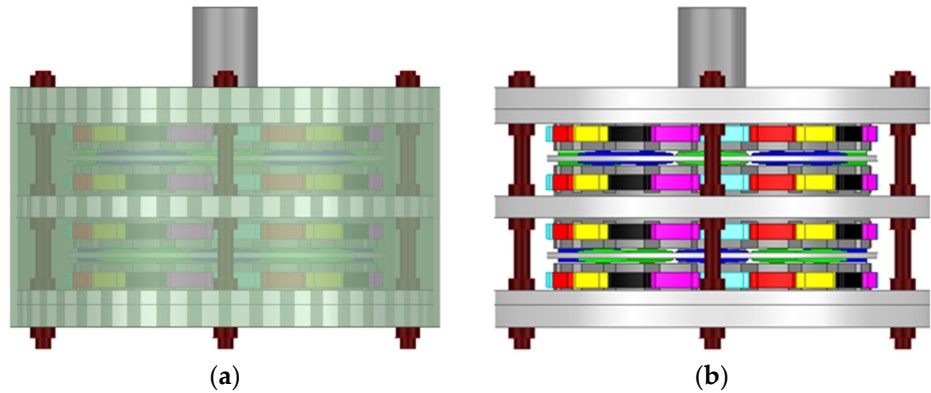

(**a**)                                    (**b**)

**Figure 23.** Realization of the proposed generator; (**a**) covered structure, (**b**) uncovered structure.

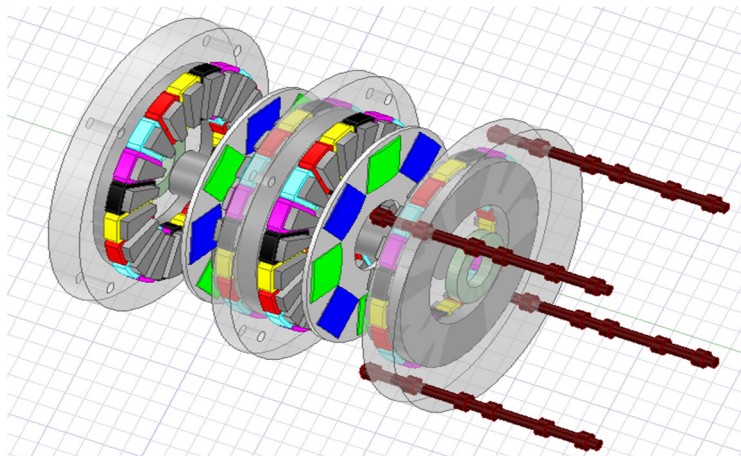

**Figure 24.** Exploded geometry of the proposed generator.

## 5. Conclusions

In this study, the process of designing the high-performance generator was described. The generator resulted from the idea of upgrading the output power ($P_{out}$) and improving the THD of the most qualified AFPMG in [10] and is also because of the rapid development of power electronic devices. By increasing the number of stator cores installed on the upper and lower sides of the original generator, $P_{out}$ of the original generator can be upgraded from 413,91 watts to 3.77 kW. By varying the pole width to pole pitch ratio ($\alpha_i$) of the arc-shaped permanent magnet and varying the slot opening width ($S_o$), an optimal rotor configuration was achieved, which resulted in THD of 1.55%. Changing the dimension of the air gap length, changing the offset angle of the stator teeth on the upper side to the stator teeth on the lower side, and changing the stator yoke height are the control factors in the Taguchi method and were validated by the Artificial Neural Network to achieve the goal of generating the lowest THD, which resulted in THD of 0.66% being realized. The improvement of 94.72% in ripple torque and the improvement of 88.78% in the ripple of the rectified voltage were reached for the proposed generator when compared to the original generator. The improvements in the ripple torque and in the ripple of the rectified voltage are proof of generating the lowest THD and adopting the five-phase system on the proposed generator. From a mechanical point of view, the geometry of the proposed generator is not too complicated. In pace with innovation within manufacturing technology, and in an effort to utilize the energy from the wind and to produce a high-performance generator, this proposed generator can be a reference for small-scale wind power generation to help societies in remote areas.

**Author Contributions:** Conceptualization, C.-Y.H. and K.W.; methodology, C.-Y.H. and K.W.; software, K.W.; validation, C.-Y.H., K.W. and N.-C.Y.; formal analysis, C.-Y.H. and K.W.; investigation, C.-Y.H., K.W. and N.-C.Y.; resources, C.-Y.H.; data curation, C.-Y.H. and K.W.; writing—original draft preparation, C.-Y.H. and K.W.; writing—review and editing, C.-Y.H., K.W. and N.-C.Y.; visualization, C.-Y.H. and K.W.; supervision, C.-Y.H. and N.-C.Y.; project administration, C.-Y.H., K.W. and N.-C.Y. All authors have read and agreed to the published version of the manuscript.

**Funding:** The research has been supported by the Ministry of Science and Technology Taiwan (MOST 109-2221-E-011-051-MY2, MOST 108-2637-E-011-002).

**Institutional Review Board Statement:** Not applicable.

**Informed Consent Statement:** Not applicable.

**Data Availability Statement:** All data generated or analyzed to support the findings of the present study are included this article. The raw data can be obtained from the authors, upon reasonable request.

**Acknowledgments:** The authors would like to thank the Editor-in-Chief, Editor, and Reviewers for their valuable reviews.

**Conflicts of Interest:** The authors declare no conflict of interest.

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
