# Peer review of "High-Performance Five-Phase Axial Flux Permanent Magnet Generator for Small-Scale Vertical Axis Wind Turbine"

_applsci, doi:10.3390/app12073632_

Round 1
Reviewer 1 Report
A general remark concerns the choice to optimize the machine by minimizing the THD: a more complete explanation and discussion of this choice should be given, considering the following aspects:
- the machine voltages are the input of a diode rectifier, that is a source of additional waveform distortion; so, which is the need to improve voltage waveform quality, considering that the generator terminals do not feed any load directly?
- some parameter changes, in particular air-gap width and offset angle, can negatively affect other important performance figures of merit (for example winding factor, efficiency): this should be carefully taken into account, besides THD.
Please, add a discussion about these aspects.
In the following, all the other specific remarks and suggestions:
- title: "Vetical-Axist" should be corrected in "Vertical-Axis";
- line 13: "becomes" instead of "become";
- line 14: "country" instead of "county";
- lines 39-41: standard winding is a three one; so, multiphase windings should be windings with number of phases 4 or higher;
- line 52: the meaning of "capacity of a capacitor" is obscure: here it is introduced without any explanation; only at the end of the paper it is clear that it is referred to a filtering capacitor at the diode bridge rectifier output;
- lines 77,78: maybe the sentence "Result and discussion are discussed in section 3 and closed with the conclusion in section 4" could be changed in "Results and discussion are reported in section 3 and completed with the conclusion in section 4";
- lines 88, 89: the sentence "The winding type is changed from core wound to tooth wound to maximize the utilization of the stator winding" should be explained;
- line 90: "modified" instead of "repaired";
- lines 106-110: the different parts of the machine are indicated with letters a, b,..., k; the names a, b, c, d, e given to the five phases are not necessary and their introduction is confusing with the parts letters;
- line 114: row 1 of table 1: "233 and 135" should be written as "135 and 233" (inner and outer diameters);
- line 119: "wounded" should be substituted by "wound" (wounded means injured);
- line 124: "phosor" should be corrected in "phasor";
- lines 131-133: the sentence "From the figure ... 4b" is not clear and should be re-written;
- figure 4b: the yellow coils are scarcely visible and should be redrawn in another color (for example: green);
- line 151,152: "in start connection" should be corrected in "in star connection";
- line 175: a sentence should be added to explain how the disposition of fig. 6 is corresponding to that shown in fig. 2: fig. 6 topology is simply consisting of two axially coupled subsystems as in fig. 6? Please, explain explicitly;
- again fig. 6: the dimension B is not clearly visible: please, displace it;
- line 198: "non-linier" should be changed in "non-linear";
- line 207: it is written three neurons, but the hidden layer 2 of fig. 7 seems consisting of just two neurons;
- lines 220-260: the paragraph is very long and it contains comments concerning several following diagrams: it could be better to insert diagrams among the comment, to make more clear the understanding;
- line 252: "combination" should be "combinations";
- fig. 10: green and red diagrams consist of just two point: is it meaningful to consider a regression with correlation equal to 1 in these cases?
- line 286: table 6: the singularity of row 8 (predicted error = 32.87%) should be more completely discussed;
- lines 295-298: the sentence is not clear and should be re-phrased;
- lines 300-318: use dotted list and insert figures among comments, to make more clear the understanding;
- line 314: even if the Maxwell numerical format gives 3765.1912 Watt, it would be better to reduce the number of digits;
- line 316: which loss items are included in the reported efficiency value (92.45%)?
- fig. 12: the caption (a) reports "symmetrical position", but the two stator sides in the upper image appear displaced between them: please, check and/or explain;
- fig. 13, 14 and 15: the diagrams show that the adopted optimization reduces distortion and torque ripple; however, a significant reduction occurs also in the fundamental voltage component, in the average torque and in the output power: please, discuss in detail pros and cons of the adopted optimization criterion;
- line 341: the result 75.6135 volt should be reported with a lower number of digits;
- fig. 17: the table corresponding to the Maxwell calculated voltage waveforms reports an average value equal to 74.3884 V that differ with respect the previous declared value: why?
Author Response
Thank your very much for revising our manuscript

Reviewer 2 Report
- It is not clear that the power density is volume ratio [W/m^3] or mass ratio [W/kg]. The authors must mention the numerical power density.
- The axial-flux machine presented in reference 1 is different in size. The authors must show the power density of the referenced machine and describe superiority of proposed machine.
- It is good results that the ripple of moving torque is significantly reduced. But the authors don’t mention the spatial harmonics. We want to know the cogging torque of the proposed machine.
- The structure of propose machine is very complicate. The authors must mention the mechanical realization of the proposed machine.
- Does the title “Vetical-Axist Wind Turbine” mean “Vertica-Axis Wind Turbine” ?
Author Response
Thanks you very much for revising our manuscript

Round 2
Reviewer 1 Report
The manuscript has been fully revised and significantly improved.
No further remarks or requests